# Effects of *Passovia ovata* Mistletoe on Pro-Inflammatory Markers In Vitro and In Vivo

**DOI:** 10.3390/plants12091814

**Published:** 2023-04-28

**Authors:** Isadora de Fátima Braga Magalhães, Ana Letícia Marinho Figueirêdo, Elizeu Mendes da Silva, Adryan Adam Batalha de Miranda, Cláudia Quintino da Rocha, Katia da Silva Calabrese, Fernando Almeida-Souza, Ana Lúcia Abreu-Silva

**Affiliations:** 1Postgraduate Program in Animal Science, State University of Maranhão, Sao Luis 65055-310, Brazil; isadoradefatimamagalhaes@gmail.com (I.d.F.B.M.); abreusilva.ana@gmail.com (A.L.A.-S.); 2Veterinary Medicine Graduation School, State University of Maranhão, Sao Luis 65055-310, Brazil; 3Postgraduate Programs in Chemistry, Federal University of Maranhão, São Luís 65080-805, Brazil; 4Laboratory of Immunomodulation and Protozoology, Oswaldo Cruz Institute, Oswaldo Cruz Foundation, Rio de Janeiro 21041-250, Brazil

**Keywords:** mistletoe, inflammation, macrophage, cytokine, mast cell

## Abstract

New agents that can suppress inflammatory responses are being sought, since chronic inflammation is associated with several pathologies. This work aims to elucidate phytochemicals from the hydroethanolic extract of mistletoe *Passovia ovata* (POH) and its anti-inflammatory potential. POH is submitted to HPLC-UV, qualitative analysis of chemical constituents, and flavonoid quantification. Cytotoxicity is evaluated in RAW 264.7 macrophages by MTT. LPS-stimulated RAW 264.7 cells are treated with POH and, after 48 h, the nitrite and cytokine levels are quantified. BALB/c mice are treated by gavage with POH and stimulated with λ-carrageenan to induce paw oedema or peritonitis. POH yield is 25% with anthraquinones, tannins, anthocyanins, anthocyanidins, flavonols, catechins and flavanones present and flavonoid content of 4.44 ± 0.157 mg QE/g dry weight. POH exhibits low cytotoxicity and significantly reduced (*p* < 0.01) nitrite, IL-1β, IL-6, and TNF-α quantification at 500 μg/mL. POH at 500 mg/kg prevents paw edema increase and also reduces inflammatory infiltrate and mast cells in the footpad. In the peritonitis model, POH does not influence cytokines levels or cell counts. Overall, POH demonstrates a high concentration of flavonoids and prominent effects in the reduction in pro-inflammatory markers in vitro and in the inhibition of paw oedema.

## 1. Introduction

Inflammation is a response to a harmful stimulus created by an internal or external agent of microbial, autoimmune, allergic, metabolic, and/or physical origin [1]. This response can range from being local and delimited to being a systemic inflammation, with a broad and complex sequence of events [2].

The inflammatory response involves leukocyte cells such as macrophages, neutrophils, and lymphocytes. These cells release pro-inflammatory cytokines, vasoactive amines, peptides, and acute phase proteins, which orchestrate inflammation. This process also demands a balance between neutralization of an agent and control to heal and restore the damaged area [3]. 

Chronic inflammation can be involved in several pathologies such as cardiovascular diseases, autoimmune and neurodegenerative diseases, and cancer [4], as pro-inflammatory cytokines, such as IL-1, IL-6, and TNF-α, are involved in the increase in platelet activity, interference of endothelial cells functions, activation of plasma coagulation cascade, reduction in fibrinolytic action, increase in platelet reactivity, and reduction in physiological anticoagulants [5]. 

Several problems have been associated with current anti-inflammatory therapy such as adverse effects and drug resistance. Therefore, new ways of controlling the inflammatory process are sought. In this context, medicinal plants emerge as a therapeutic alternative, and although popular use may suggest potentially important species, research is essential to prove their pharmacological effect [6].

Loranthaceae and Santalaceae form two of the largest families of mistletoes [7], a group of hemiparasitic plants that through roots named haustorium obtain water and minerals from the host plant [8]. The genus *Passovia* represents neotropical mistletoes from the Loranthacea family [9], a group that has been recognized for many years for its therapeutic properties [10]. *Passovia ovata* is a species with ovate leaves, apex usually acute or attenuated, and little-branched terminals [9]. It can be considered a generalist species, as it tends to have a wider geographic distribution. This species is a facultative aluminum accumulator plant; therefore, regardless of whether the host plant accumulates aluminum or not, *P. ovata* will not show symptoms of deficiency or toxicity [11].

To our knowledge, no previous study has been reported on the anti-inflammatory activity of *P. ovata*. Therefore, we investigated the phytochemical and immunomodulatory properties of the hydroethanolic extract from leaves of *P. ovata* using cultured RAW 264.7 cells and murine in vivo models.

## 2. Results

### 2.1. Yield and Physicochemical Properties

The extraction yield of POH was 25%. POH was submitted for analysis by high-performance liquid chromatography photodiode array detection method (HPLC-UV) (Figure 1).

HPLC-UV chromatogram revealed a significant presence in the content of major constituents phenólicos with absorbance in 254 nm and the phytochemical screening results of POH (Table 1) reveal the presence of anthraquinones, tannins, anthocyanins, anthocyanidins, flavonols, catechins, and flavanones. The total flavonoid content in POH was 4.44 ± 0.157 mg QE/g dry weight.

### 2.2. Effects of the Hydroethanolic Extract of P. ovata on Macrophages RAW 264.7

The effect of POH on the viability of RAW 264.7 macrophages (Figure 2).

After testing by MTT, POH demonstrated low cytotoxicity as it did not reduce cell viability in macrophages RAW 264.7 submitted to the same treatment (125, 250, and 500 μg/mL) used in the other in vitro assays. Cytotoxicity also did not increase with increasing treatment time. In LPS-stimulated RAW 264.7 macrophages, POH concentration of 500 μg/mL significatively (*p* < 0.01) reduced nitrite (Figure 3A), IL-1β (Figure 3B), IL-6 (Figure 3C), and TNF-α (Figure 3D). The same was observed in cells treated with dexamethasone at 5 μg/mL. The quantification of nitrite and cytokines in the group stimulated with LPS was significantly (*p* < 0.05) higher when compared to the control group (unstimulated and untreated).

### 2.3. In Vivo Anti-Inflammatory Effect of POH

#### 2.3.1. Paw Edema

Edema thickness was used to evaluate the effect of POH treatment. Edema thickness was lower than the PBS group after two, three, and four hours of treatment with dexamethasone, and after two and four hours of treatment with POH at 250 mg/kg and 500 mg/kg doses (Figure 4). Table 2 also shows the effect of POH or standard drug dexamethasone in the reduction in paw edema in stimulated BALB/c mice after one, two, three, and four hours of treatment. The subcutaneous administration of λ-carrageenan caused an increase in footpad thickness and some treatments significantly prevented the edema: POH administered at a dose of 500 mg/kg (*p* < 0.01 and *p* < 0.05) or 250 mg/kg (*p* < 0.05) and dexamethasone at 5 mg/kg dose (*p* < 0.0001).

Footpad histological images of tissue fragments stained with hematoxylin and eosin demonstrated a greater tissue preservation in the non-stimulated group (Figure 5A) and in groups treated with 500 mg/kg of POH (Figure 5E) and dexamethasone (Figure 5F) than in the untreated and non-stimulated group. A higher inflammatory infiltrate was observed in the stimulated and PBS-treated group (Figure 5B) and in the group treated with the lowest dose (125 mg/kg) of POH (Figure 5C).

After histological microscopic analysis of footpad tissue, it was observed that inflammatory infiltrate was present in all groups of animals used in the experiment, being discreet (+) in unstimulated and untreated group, intense (+++) in group stimulated and treated with PBS, and intense (+++), moderate (++), and discrete (+) in groups treated with 125, 250, and 500 mg/kg of POH and 5 mg/kg of dexamethasone, respectively (Table 3).

In mast cells from histological footpad slides stained with toluidine blue (Figure 6), the stimulated and PBS-treated group showed a significant increase (*p* < 0.001) in the number of mast cells when compared to the non-stimulated and untreated group. The groups treated with 500 mg/kg of POH, or dexamethasone (5 mg/kg) had a significant reduction, *p* < 0.001 and *p* < 0.0001, respectively, in mast cells, when compared to the stimulated and PBS-treated control (Figure 6G).

#### 2.3.2. λ-Carrageenan-Induced Peritonitis

As shown in Figure 7A, the induction of peritonitis by carrageenan stimulated a significant (*p* < 0.01) increase in cell migration to the peritoneal cavity when compared to the untreated group (0.957 × 10^7^ cells/mL vs. 3.504 × 10^6^ cells/mL). The only group that showed a significant reduction in cell count (Figure 7A, *p* < 0.01), IL-1β (Figure 7B, *p* < 0.05), and IL-6 (Figure 7C, *p* < 0.01) quantification in serum was the one treated with dexamethasone (5 mg/kg) (0.594 × 10^7^ cells/mL). The hematological evaluation consisted of red blood cell count (RBC), hemoglobin concentration (HGB), hematocrit (HCT), mean corpuscular volume (MCV), mean corpuscular hemoglobin concentration (MCHC), leukocyte counts (WBC), and platelet counts (PLT). No statistical alterations were observed between the groups, as seen in Table 4.

## 3. Discussion

*P. ovata* is a neotropical mistletoe that has not been explored for its pharmacological potential, but studies reporting the promising effects of other mistletoes led us to explore its chemical compounds and immunomodulatory effect.

To better evaluate the composition, POH analysis was performed through HPLC, a widely used tool to evaluate plant origin extracts sensitively and accurately as a prerequisite for its standardization [12].

The compounds found in the hydroethanolic extract (anthraquinones, tannins, flavonols, and catechins) have proven antitumor, laxative, antimicrobial [13,14,15], antiviral, antioxidant, and anti-inflammatory action [16,17], which may be directly related to the therapeutic potential of *P. ovata*.

The total flavonoids content in POH was higher than the ones found in other mistletoes in similar assays such as the alcoholic extract from leaves of *Phthirusa* sp, (10.51 ± 1.81 mg QE/g) [18] and methanolic extract from *Dendrophthoe pentandra* (L.) Miq. (4.25 ± 21.21 mg EQ/g) [19]. Flavonoids are produced in plants to help fight oxidative stress and act as growth regulators. Many flavonoids isolated from plant extracts have shown anti-inflammatory effect, being, therefore, an important parameter for the elaboration of new drugs against inflammatory disorders [20].

In the in vitro assay, POH showed low cytotoxicity in RAW 264.7 macrophages after 24 and 48 h treatment, encouraging their possible future application. To evaluate the potential anti-inflammatory effect of POH, macrophages were stimulated with LPS, and cytokines and nitrite were quantified.

Macrophages form a heterogeneous cell population that can be considered classically activated by LPS and/or Th1 cytokines, which generates a pro-inflammatory profile named M1. Macrophages M1 trigger a high production of cytokines such as TNF-α, IL-12, IL-6, and IL-1β and higher amounts of nitric oxide (NO). Alternative activation is stimulated by Th2 cytokines such as IL-13 and IL-4 and defines a population with immunoregulatory and anti-inflammatory functions marked by the production of cytokines such as TGF-β and IL-10. While the M1 profile encourages the elimination of the stimulus, the M2 profile seeks to limit inflammation and promote tissue recovery and remodeling [21].

In the evaluation of potentially therapeutic compounds in macrophages, the measurement of nitrite is an important parameter as it represents an indirect way of measuring NO, a gas with signaling, regulatory, and microbicidal functions. It is produced by macrophages under circumstances that stimulate inflammation and induce them to a M1 profile [22]. After 48 h treatment with POH at 500 μg/mL, there was observed a nitrite quantification reduction in LPS-stimulated macrophages. Previous studies indicated that hydroethanolic extract from *S. vulgaris* mistletoe at 80 and 160 μg/mL concentrations significantly reduced the production of NO in the J774 macrophages strain [23], and the methanolic fraction from leaves of *Phragmanthera capitata* at 25 μg/mL concentration also reduced nitrite levels [24].

Macrophages are among the main sources of cytokines when exposed to factors that stimulate the immune response [25]. LPS is an endotoxin that induces macrophage activation, formation of self-amplifying loops, and cytokine production [26] through Toll-like receptors (TLR) [27].

In the evaluation of plant-derived compounds, IL1-β, IL-6, and TNF-α are among the main mediators used as parameters for evaluating possibly anti-inflammatory cellular responses [28]. Here, the same concentration of POH (500 μg/mL) that inhibited NO production also reduced the quantification of IL-1β, IL-6, and TNF-α. These cytokines are considered pro-inflammatory and are liberated when a stimulus, such as LPS from Gram-negative bacteria, activates TLR4 [29]. Macrophages isolated from BALB/c mice when treated with *Viscum album coloratum*, known as Korean mistletoe, also showed reduction on the quantification of these cytokines [30].

In the paw edema test, parameters related to the reduction in edema induced by the phlogogen agent and through the histological analysis of the inflammatory infiltrate and mast cell count of footpad tissue were evaluated. Mast cells are modulator cells of the immune system that act in acute inflammatory responses, such as those that occur in allergic reactions and in the elimination of pathogens [31]. They are among the first cells to act against antigens and initiate local inflammation mechanisms [32]. The model of inflammation induced by carrageenan involves an early phase in which mast cells release histamine and a later phase in which there is neutrophil migration and increased production and release of cytokines and prostaglandin [33].

In our work, the concentration of 500 mg/kg of POH induced the bigger reduction in paw edema, as well as a reduction in mast cells, events that may be related with the activation and release of granules and with mediators such as histamine that increase the permeability of the vessels and consequently cause the release of liquid into the extracellular environment, promoting local edema [34]. In a trial with another mistletoe, also of the Loranthacea family, *S. vulgaris*, the ethanol leaf extract reduced edema at 50 and 100 mg/kg doses [23].

Despite the results in the paw edema test, in the peritonitis model in BALB/c mice, treatment with POH did not demonstrate a reduction in the cell count of the intraperitoneal lavage and in cytokines IL-6 and IL-1β, although a visible trend in the reduction in cytokines is perceptible as the treatment dose increases. In these assays, only the standard drug dexamethasone showed a reduction effect in cell count and in cytokines IL-6 and IL-1β quantification.

The inflammatory process is highly complex and involves several cell types and at least 15 types of cytokines, adrenocorticotrophic hormones, and acute phase proteins, which manifest as features of redness, edema, heat, and pain [35]. Therefore, our work is a preliminary trial on the effect of this mistletoe on pro-inflammatory markers, but further elucidation of the effects of derived compounds from *P. ovata* on the immune system will be better determined in future trials.

## 4. Materials and Methods

### 4.1. Plant Material and Hydroethanolic Extract Preparation

The leaves and stems of *P. ovata* (Pohle x DC.) Tiegh. were collected in July 2019 at the Farm School of the State University of Maranhão–UEMA, located in the city of São Luís, Brazil, where the plant was parasitizing a mango tree (*Mangifera indica*). A voucher specimen was deposited with the number 5516 and identified by the Herbarium Rosa Mochel, UEMA. The genetic heritage material access was granted by the National System for the Management of Genetic Heritage and Associated Traditional Knowledge (Sistema Nacional de Gestão do Patrimônio Genético e do Conhecimento Tradicional Associado–SisGen) with reference record AFC60DB. *P. ovata* hydroethanolic extract (POH) was obtained by maceration (20% *w*/*v*) during a period of 12 days, with solvent (ethanol 70%, Êxodo Científica, Hortolândia, Brazil) changes every three days. The final extract was concentrated in a rotary evaporator (Fisatom 802, São Paulo, Brazil) and lyophilized at −70 °C (Terroni Enterprise, São Carlos, Brazil). The yield of the POH was calculated according to the following formula [36]:$$Total\;yield\;\left(\%\right)=\frac{mass\;of\;the\;extract\;in\;grams}{mass\;of\;the\;dried\;plant\;material\;in\;grams}\times 100$$

### 4.2. High Performance Liquid Chromatography (HPLC-UV)

POH was analyzed using the high-performance liquid chromatography photodiode array detection method (HPLC-UV) using a 254 nm wavelength (Shimadzu, Kyoto, Japan). Thus, 10 mg of POH were added to 800 μL of methanol (Êxodo Científica, Hortolândia, BRA) HPLC and 200 μL of distilled water for homogenization in a sonicator (Unique, São Paulo, Brazil). The gradient exploratory consisted of 5–100% methanol (Êxodo Científica, Hortolândia, Brazil) in 40 min and 100–100% from 40 to 50 min on photo-diode-array detection (PDA) (Shimadzu, Kyoto, Japan). Then, 20 μL of POH was injected into the HPCL, and the peaks were shown on a chromatogram.

### 4.3. Phytochemical Screening by Qualitative Analysis

To identify the main classes of secondary metabolites, a qualitative phytochemical analysis was performed. The tests were based on color change, precipitate formation, or foam after the addition of reagents, and resulted in the analysis of the presence of anthraquinones, saponins, tannins, anthocyanidins, chalcones, leucoanthocyanidin, catechins, flavanones, flavones, flavanols, xanthones, aurones, flavonols, and coumarins [37,38].

#### 4.3.1. Anthraquinones

In a 96-well plate (Jet Biofil, Guangzhou, China), 150 µL of POH at 20 mg/mL in methanol (Êxodo Científica, Hortolândia, Brazil) was added to 50 µL of 0.5 M sodium hydroxide (NaOH) (Sigma-Aldrich, St. Louis, MO, USA). The appearance of a red color indicates the presence of anthraquinones.

#### 4.3.2. Saponins

POH at 20 mg/mL in methanol (Êxodo Científica, Hortolândia, Brazil) was added to 2 mL of distilled water in a test tube. The tube was then shaken vigorously for 30 s and allowed to stand for 10 min. The presence of foam with a height greater than 1 cm indicates the presence of saponins.

#### 4.3.3. Tannins

POH (20 mg/mL) was placed in a test tube and 2.5% gelatin (Sigma-Aldrich, St. Louis, MO, USA) was added dropwise. The formation of a white precipitate indicates the presence of tannins.

#### 4.3.4. Coumarins

The determination of coumarin was performed with the addition of approximately 1.5 cm in diameter of POH on a piece of filter paper (Merck, Rio de Janeiro, Brazil). After drying, a drop of aqueous potassium hydroxide (KOH) (Sigma-Aldrich, St. Louis, MO, USA) solution was added to each point and the filter paper was placed under ultraviolet (UV) light using a Spectroline UV (Spectronic Corporation, Westburg, NY, USA) for three minutes. The observation of a strong and clearly visible bluish fluorescence in the sample is indicative of a coumarin presence.

#### 4.3.5. Anthocyanins, Anthocyanidins, Flavone, Flavanols, Xanthones, Chalcones, Aurones, Leukoanthocyanidins, Catechins, Flavonols, and Flavanones

POH (20 mg/mL) was added to test tubes. In tube 1, 0.5 M hydrochloric acid (HCl) (Sigma-Aldrich, St. Louis, MO, USA) was added until reaching a pH of 3.0. In the second and third tubes, 0.5 M NaOH (Sigma-Aldrich, St. Louis, MO, USA) was added until reaching pH of 8.0 and 11.0, respectively. The appearance of specific colors indicates the presence of these compounds. The red, lilac, and blue-purple colors in tubes 1, 2, and 3, respectively, indicate the presence of anthocyanins and anthocyanidins. The yellow color in tube 3 indicates the presence of flavone, flavanols, and xanthones. The red and red-purple colors in tubes 1 and 3, respectively, indicate the presence of chalcones and aurones. The presence of the red color in tube 1 indicates the presence of leukoanthocyanidins. The presence of a yellowish-brown color indicates the presence of catechins. In addition, the presence of the red-orange color in tube 3 indicates the presence of flavonols and flavanones.

### 4.4. Flavonoid Quantification

The quantification of the flavonoid content was performed in a 96-well microplate (Jet Biofil, Guangzhou, China). In each well, 200 μL of POH (1 mg/mL) and 100 μL of 2% aluminum chloride (AlCl_3_) (Êxodo Científica, Hortolândia, Brazil) in methanol (Êxodo Científica, Hortolândia, Brazil) were added. After incubation in the dark for 30 min, the reading was performed in a spectrophotometer (Biochrom Ltd., Cambridgeshire, UK) at 420 nm, and the obtained absorbances were interpolated from the quercetin (Sigma-Aldrich, St. Louis, MO, USA) calibration curve. All tested solutions were made in triplicate and the experiment was repeated twice. Blank samples were made with 300 μL of POH. Results were expressed in QE (quercetin equivalents) in mg per g of plant sample [39].

### 4.5. RAW 264.7 Macrophages Cell Culture

The cells were kept in culture flasks at 37 °C and 5% CO_2_. The medium used was Roswell Park Memorial Institute (RPMI 1640, Gibco, Gainthersburg, MD, USA) supplemented with 10% inactivated fetal bovine serum (FBS) (Gibco, Gainthersburg, MD, USA), L-glutamine (20 mM) (Gibco, Gainthersburg, MD, USA), and antibiotic solution containing penicillin (100 U/mL) and streptomycin (100 μg/mL) (Sigma-Aldrich, St. Louis, MO, USA). Cells were monitored daily under an inverted microscope, and at 80–90% confluence, the passages were performed.

### 4.6. Cytotoxicity of the Hydroethanolic Extract of P. ovata on RAW 264.7 Macrophages

The evaluation of cytotoxicity was performed according to the adaptation of the viability assay using 3-4,5-dimethyl-thiazol-2-yl-2,5-diphenyltetrazolium bromide (MTT) [40]. Cells were added at 2 × 10^6^/mL in 100 µL of RPMI 1640 medium (Gibco, Gainthersburg, MD, USA), supplemented with 10% FBS and incubated overnight at 37 °C with 5% CO_2_. After this period, the medium was removed, and cells were treated with concentrations of 1000 to 1.9 µg/mL of POH diluted in RPMI medium (Gibco, Gainthersburg, MD, USA). Two and a half hours before completing each treatment time (24 or 48 h), 10 µL of 5 mg/mL MTT (Sigma-Aldrich, St. Louis, MO, USA) was added with further incubation until completion of treatment. Subsequently, the cell medium was aspirated, and 100 µL of dimethyl sulfoxide (DMSO at 99.5%; Sigma-Aldrich, St. Louis, MO, USA) was added. The negative control consisted of medium-only wells, and the positive control consisted of cells and 1% DMSO. The absorbance values were acquired in a spectrophotometer (Biochrom Ltd., Cambridge, UK) with a wavelength of 570 nm, and a dose response curve was made to determinate the concentration that inhibits 50% of the cells (IC_50_).

### 4.7. Anti-Inflammatory Effect of the Hydroethanolic Extract of P. ovata in Lipopolysaccharide-Induced RAW 264.7 Macrophages

RAW 264.7 cells were plated on 24-well plates (Jet Biofil, Guangzhou, China) at a density of 2 × 10^6^ cells/mL per well overnight. After that, they were treated with POH, and after 1 h, stimulated groups were treated with lipopolysaccharide (LPS) (10 µg/mL) from *Escherichia coli* (Sigma-Aldrich, St. Louis, MO, USA). This assay consisted of six different groups: (1) negative control: non-LPS induced cells; (2) positive control: non-treated and LPS-stimulated cells; (3) drug reference group: treated with dexamethasone (5 μg/mL) (Sigma-Aldrich, St. Louis, MO, USA) and LPS stimulated; (4) group treated with 500 μg/mL of POH and LPS stimulated; (5) group treated with 250 μg/mL of POH and LPS stimulated; and (6) group treated with 125 μg/mL of POH and LPS stimulated. After the total incubation period of 48 h, the supernatant was collected for nitrite and IL-1β, IL-6, and TNF-α cytokines quantification.

### 4.8. In Vivo Anti-Inflammatory Activity of Hydroethanolic Extract of P. ovata

#### 4.8.1. Animals

Four-week-old male BALB/c mice weighing 20–22 g were obtained from the Institute of Science and Technology in Biomodels, Fiocruz, Rio de Janeiro-Brazil. The animals were kept in individually ventilated cages (IVC), five animals in each cage, and maintained in pathogen-free, temperature-controlled conditions, with a light cycle of 12 h light/dark cycle, at 25 ± 2 °C, and with relative humidity in a range of 55–65%. The animals were fed with a standard diet and water ad libitum. The experiments were approved by the Ethics Committee on the Use of Animals of the Instituto Oswaldo Cruz/Fiocruz (CEUA-IOC)–Rio de Janeiro (CEUA L-53/2016-A3) and conducted in accordance with the National Council for the Control of Animal Experimentation (CONCEA).

#### 4.8.2. Paw Edema Model

The evaluation of anti-inflammatory activity was performed on a model of paw edema induced by λ-carrageenan [33,41]. Mice were separated into six groups with five animals each: (1) negative control group: animals treated with PBS by gavage without stimulation; (2) positive control group: animals treated with PBS by gavage and stimulated with λ-carrageenan; (3) standard drug group: animals treated with dexamethasone (5 mg/kg) intramuscular and stimulated with λ-carrageenan; (4) animals treated with 500 mg/kg of POH by gavage and stimulated with λ-carrageenan; (5) animals treated with 250 mg/kg of POH by gavage and stimulated with λ-carrageenan; and (6) animals treated with 125 mg/kg of POH by gavage and stimulated with λ-carrageenan. One hour after treatment, the edema was induced with 25 μL of 1% λ-carrageenan in phosphate-buffered saline (PBS) inoculated via sub-plantar tissue of the right hind paw, except for group 1, which received only 25 μL of PBS in the right hind paw. After 1, 2, 3, and 4 h from stimulation, pad swelling at both paws was measured using a Schnelltaster dial gauge caliper (Kröplin GmbH, Hessen, Germany). The animals were euthanized with ketamine (100 mg/kg, Syntec, Barueri, Brazil) and xylazine (20 mg/kg, Syntec, Barueri, Brazil) in a 1:1 ratio intraperitoneally. The difference of thickness between the inoculated pad and the non-inoculated pad was determined and the inhibition of edema in percentage was calculated.

Paw edema tissue fragments were collected for histological analysis of the inflammatory tissue. Tissue fragments were fixed for 48 h in 10% formaldehyde (Dinâmica Química Contemporânea, Indaiatuba, Brazil) solution. Then they were dehydrated, embedded in pure paraffin (Solven^®^, Hortolandia, Brazil), and sectioned at 5 μm in microtome (LUPE MRP03, LupeTec, São Carlos, Brazil). Tissues were submitted to Hematoxylin (Alphatec, Rio de Janeiro, Brazil) and Eosin (Isofar, Rio de Janeiro, Brazil) staining for analysis of the inflammatory infiltrate under light microscopy (Leica Microsystems, Mannheim, Germany) at 40× magnification. Representative areas were selected in five fields and a qualitative scale was used to classify the infiltrate: (+): discreet; (++): moderate; and (+++): intense. For the quantification of mast cells in the inflammatory infiltrate, toluidine blue (Alphatec, Rio de Janeiro, Brazil) staining was used. Representative areas were selected in five fields for counting under a light microscope with an approximate 40× magnification (Leica Microsystems, Mannheim, Germany). Purple-stained cells (interpreted as mast cells) were counted.

#### 4.8.3. λ-Carrageenan-Induced Peritonitis

The inflammatory peritonitis model was also performed [42]. Stimulated groups were inoculated with 250 μL of 1% λ-carrageenan intraperitoneally, or 250 μL of PBS (negative control), 1 h after treatment. The experiment consisted of six groups with five animals each, divided the same way as the paw edema model. Four hours after λ-carrageenan stimulation, mice were euthanized with a 1:1 mixture of ketamine (100 mg/kg, Syntec, São Paulo, Brazil) and xylazine (20 mg/kg, Syntec, São Paulo, Brazil) administered intramuscularly. Blood was collected through intracardiac puncture for ELISA cytokines IL-1β and IL-6 measurement from the serum. Intraperitoneal lavage with PBS was also performed for further total leukocyte count in the Neubauer chamber.

### 4.9. Nitrite and Cytokines Quantification

Nitric oxide (NO) evaluation was indirectly determined by nitrite concentration by the Griess colorimetric method [43]. Thus, 50 μL of supernatant from each well was added to 50 μL of Griess reagent (1:1 mixture of a 1% solution of sulfanilamide (Sigma-Aldrich, St. Louis, MO, USA) in 2.5% orthophosphoric acid (H_3_PO_4_) (Sigma-Aldrich, St. Louis, MO, USA) and 0.1% solution of N-(1-naphthyl)-ethylenediamine dihydrochloride (Sigma-Aldrich, St. Louis, MO, USA) in a 96-well microplate (Jet Biofil, Guangzhou, China). After 10 min, the optical density (absorbance) was spectrophotometrically (Biochrom, Cambridge, UK) evaluated by ELISA reader with length waveform of 570 nm. Results were expressed in μM of sodium nitrate (NaNO_2_) (Sigma-Aldrich, St. Louis, MO, USA), based on a standard reference curve. Cytokines IL-1β (Cat. No. 3259705), TNF-α (Cat. No. 9008894), and IL-6 (Cat. No. 7333548) were quantified by a BD OptEIA™ enzyme-linked immunosorbent assay (ELISA) test (BD Biosciences, San Jose, CA, USA) following the manufacturer’s instructions.

### 4.10. Statistical Analysis

Results were represented as mean ± standard deviation and analyzed by a Kruskal–Wallis test with Dunn’s post-test or by two-way ANOVA with Bonferroni comparisons in the Graph Pad 9.0 software package (GraphPad Software, San Diego, CA, USA). Significant differences were considered as *p* < 0.05.

## 5. Conclusions

Mistletoes are well-known for their immunomodulatory potential mechanisms. Here, we developed an in vitro and in vivo study on a yet-unexplored species of neotropical mistletoe named *P. ovata*. The phytochemical profile showed a plant rich in secondary chemical compounds with high pharmacological potential, in addition to high quantification of flavonoids. The hydroethanolic extract from the leaves of *P. ovata* demonstrated low cytotoxicity and ability to reduce nitrite and cytokines considered pro-inflammatory at our highest tested dose (500 μg/mL). In vivo assays with BALB/c mice showed that the oral treatment with a higher dose of the extract (500 mg/kg) reduced mast cell migration and paw-stimulated edema, but in another in vivo inflammatory model with peritonitis in BALB/c mice, treatment with the extract did not influence intraperitoneal lavage cell counts and serum cytokine concentrations.

In general, we conclude that the hydroethanolic extract of *P. ovata* leaves is rich in secondary chemical compounds with a high concentration of flavonoids, has immunomodulatory effect in vitro, and can inhibit the formation of paw edema and mast cell count in the paw tissue of mice. This is the first report on the therapeutic effects of *P. ovata*; therefore, we encourage that, in further studies, the pathways involved in the immunomodulatory effect, as well as the use of other inflammatory models and isolation of active compounds, are performed.

## Figures and Tables

**Figure 1 plants-12-01814-f001:**
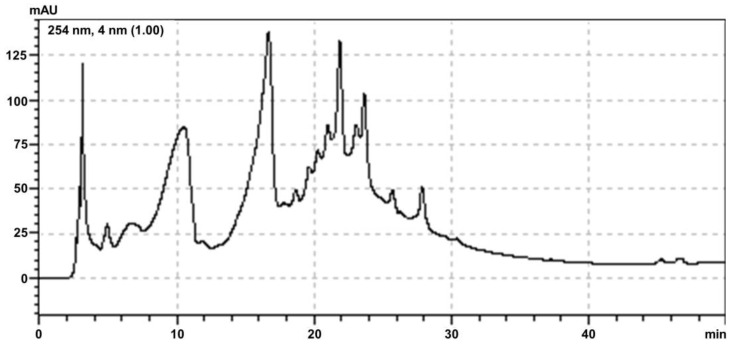
HPLC-UV chromatogram of hydroethanolic extract from *Passovia ovata* leaves at 254 nm.

**Figure 2 plants-12-01814-f002:**
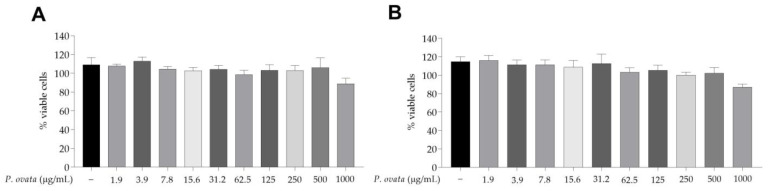
*Passovia ovata* hydroethanolic extract in vitro toxicity in RAW 264.7 macrophages. Cell viability percentages obtained by the MTT assay after 24 h (**A**) and 48 h (**B**) from treatment with concentrations from 1000 to 1.9 μg/mL of the extract. Control group received only RPMI 1640 medium. Data represent the mean ± standard for three different experiments performed in triplicate. Abbreviation: microgram per milliliter (µg/mL).

**Figure 3 plants-12-01814-f003:**
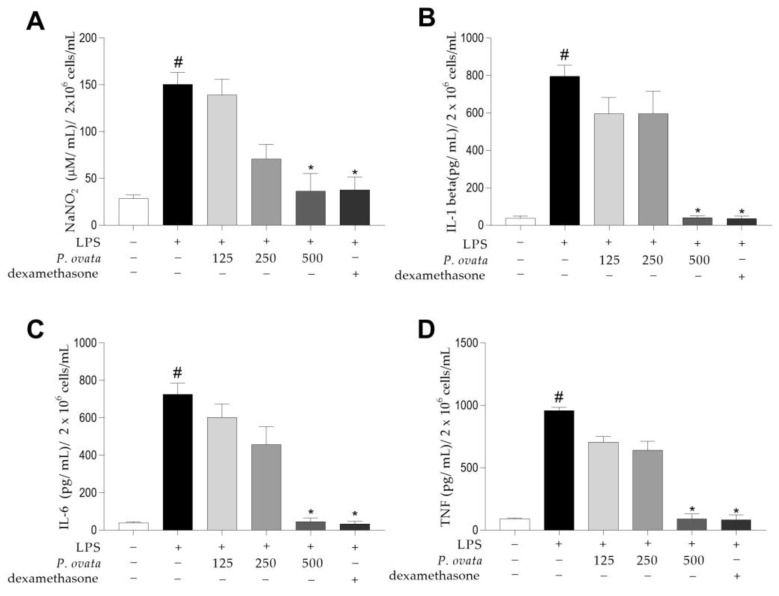
Quantification of nitrite and cytokines in culture supernatant of RAW 264.7 cells treated with 125, 250, and 500 mg/kg concentrations of hydroethanolic extract of *Passovia ovata* leaves or dexamethasone (5 μg/mL) and stimulated by LPS (10 µg/mL) for 48 h. (**A**) Nitrite, (**B**) IL-1 β, (**C**) IL-6, and (**D**) TNF-α levels. Results are expressed as the mean ± SD of three different experiments performed in triplicate. # *p* < 0.01 comparing the untreated and unstimulated group; * *p* < 0.05, when compared with the group stimulated with LPS, after Kruskal–Wallis test and Dunn’s multiple comparison analysis. Abbreviations: lipopolysaccharide (LPS); sodium nitrite (NaNO_2_); interleukin-6 (IL-6); interleukin-1 beta (IL-1β); tumor necrosis factor alpha (TNF-α); micromole per milliliter (μM/mL); cells/mL (number of cells per milliliter); pg/mL (picograms per milliliter).

**Figure 4 plants-12-01814-f004:**
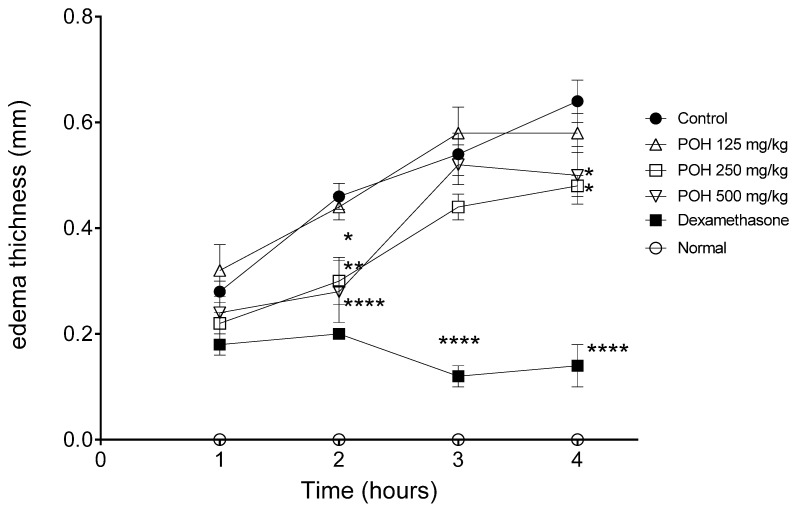
Kinetic of edema thickness in millimeters of paw edema induced by 1% λ-carrageenan in BALB/c mice treated with hydroethanolic extract of *Passovia ovata* at 125, 250, or 500 mg/kg or dexamethasone at 5 mg/kg. Normal group is representative of untreated and unstimulated group and control is representative of the group induced with λ-carrageenan and treated with PBS. Data represent mean ± standard deviation of experiment of three different experiments performed in quintuplicate * *p* < 0.05; ** *p* < 0.01; **** *p* < 0.0001, after analysis of variance (two-way ANOVA) and Bonferroni’s multiple comparisons test when compared with the control group induced with λ-carrageenan and treated with PBS. Abbreviations: *P. ovata* hydroethanolic extract (POH); dexamethasone (dexa), PBS (phosphate-buffered saline).

**Figure 5 plants-12-01814-f005:**
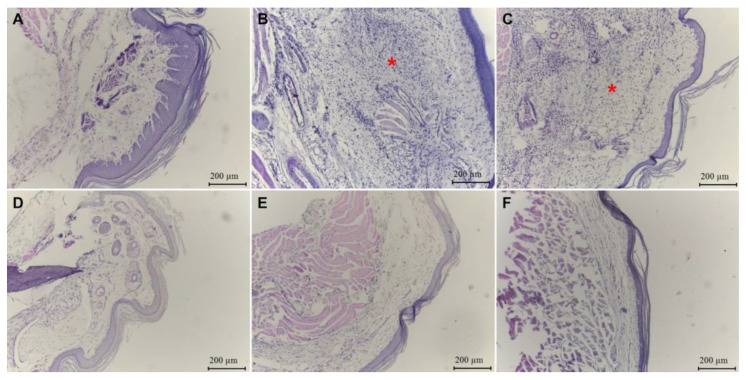
Histological analysis by staining with hematoxylin and eosin of paw edema from BALB/c mice inoculated with λ-carrageenan and treated with *Passovia ovata* hydroethanolic extract (POH). (**A**) Unstimulated and untreated group; (**B**) group inoculated with λ-carrageenan and treated with PBS solution; (**C**) group treated with 125 mg/kg of POH and inoculated with λ-carrageenan; (**D**) group treated with 250 mg/kg of POH and inoculated with λ-carrageenan; (**E**) group treated with 500 mg/kg of POH and inoculated with λ-carrageenan; (**F**) treated with dexamethasone (5 mg/kg) and inoculated with λ-carrageenan. Images are representative of three independent experiments carried out in quintuplicate. Red asterisk: inflammatory infiltrate.

**Figure 6 plants-12-01814-f006:**
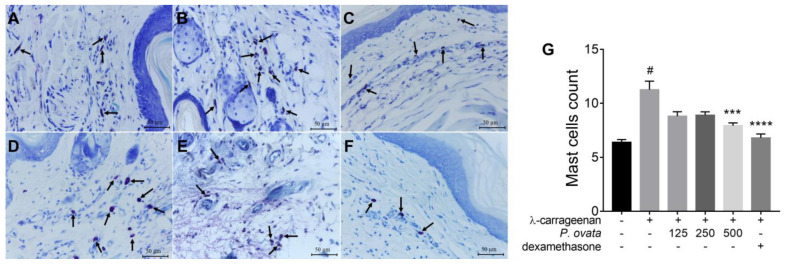
Paw edema of BALB/c mice inoculated with λ-carrageenan and treated with *Passovia ovata* hydroethanolic extract (POH). Histological analysis by staining with toluidine blue in group (**A**) unstimulated and untreated group; (**B**) inoculated with λ-carrageenan and treated with PBS solution; (**C**) treated with 125 mg/kg of POH and inoculated with λ-carrageenan; (**D**) treated with 250 mg/kg of POH and inoculated with λ-carrageenan; (**E**) treated with 500 mg/kg of POH and inoculated with λ-carrageenan; (**F**) treated with dexamethasone (5 mg/kg) and inoculated with λ-carrageenan. (**G**) Quantification of mast cells in these groups. Arrows shows mast cells. Data are expressed as mean ± standard deviation and images are representative of three independent experiments carried out in quintuplicate. # *p* < 0.001 compared with control group; *** *p* < 0.001, **** *p* < 0.0001 compared with PBS group, after analysis of variance (two-way ANOVA) followed by Kruskal–Wallis followed by Dunn’s multiple. Abbreviations: lambda carrageenan (λ-carrageenan. The sign + or – represents group submitted or not to λ-carrageenan inoculation.

**Figure 7 plants-12-01814-f007:**
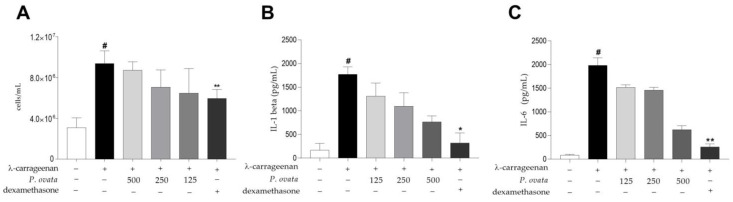
Peritonitis induced by λ-carrageenan in BALB/c mice treated with hydroethanolic extract of *Passovia ovata*. Total count of cells in intraperitoneal lavage as cells/mL (**A**), and cytokine IL-1β (**B**) and IL-6 (**C**) quantification in sera as pg/mL of BALB/c mice after four hours of peritonitis induced by λ-carrageenan 1% orally treated with 500, 250, and 125 mg/kg of P. ovata hydroalcoholic extract or dexamethasone at 5 mg/kg. The data are expressed as mean ± standard deviation and are representative of three experiments performed in quintuplicate. # *p* < 0.001 compared with unstimulated group; * *p* < 0.05, ** *p* < 0.01 compared with the group stimulated with λ-carrageenan only after Kruskal–Wallis analysis followed by Dunn’s multiple comparisons test. Abbreviations: number of cells per milliliter (cells/mL); picograms per milliliter (pg/mL); interleukin-1 beta (IL-1β); interleukin-6 (IL-6); lambda carrageenan (λ-carrageenan).

**Table 1 plants-12-01814-t001:** Classes of secondary metabolites and the respective chemical reagents used for their identification in *Passovia ovata* hydroethanolic extract. (−) absence; (+) poor; (++) moderate; (+++) abundant.

Classes of Secondary Metabolites	Chemical Reagents	Result
Anthraquinones	NaOH	+++
Saponins	Foam index	−
Tannins	Gelatin	+++
Coumarins	KOH	−
Anthocyanins and anthocyanidins	NaOH	−
Flavone, flavanols and xanthones	NaOH	−
Chalcones and Aurones	NaOH	−
Leukoanthocyanidins	HCl	−
Catechins	HCl	+++
Flavanols and flavanones	NaOH	+++

Abbreviations: sodium hydroxide (NaOH); potassium hydroxide (KOH); hydrochloric acid (HCl).

**Table 2 plants-12-01814-t002:** Thickness in millimeters and percentage inhibition of paw edema induced by 1% λ-carrageenan in BALB/c mice treated with hydroethanolic extract of *Passovia ovata* or dexamethasone.

Group	λ-Carrageenan	Dose (mg/kg)	Administration Time (% Inhibition of Edema)			
			1 h	2 h	3 h	4 h
Normal	PBS	PBS	0.014 ± 0.03	0.014 ± 0.03	0.014 ± 0.03	0.014 ± 0.03
Control	+	PBS	0.28 ± 0.045	0.46 ± 0.054	0.54 ± 0.089	0.64 ± 0.089
POH	+	125	0.32 ± 0.109	0.44 ± 0.054	0.58 ± 0.109	0.58 ± 0.083
	+	250	0.22 ± 0.044	0.3 ± 0.1 (34.78) *	0.44 ± 0.054	0.48 ± 0.044 (25.0) *
	+	500	0.24 ± 0.089	0.28 ± 0.13 (39.13) **	0.52 ± 0.083	0.5 ± 0.122 (21.87) *
Dexa	+	5	0.18 ± 0.044	0.02 ± 0.0 (56.52) ****	0.12 ± 0.044 (77.77) ****	0.14 ± 0.089 (78.12) ****

Data represent mean ± standard deviation of three different experiments performed in triplicate. * *p* < 0.05; ** *p* < 0.01; **** *p* < 0.0001, after analysis of variance (two-way ANOVA) and Bonferroni’s multiple comparisons test when compared with the control group induced with λ-carrageenan and treated with PBS. Abbreviations: *P. ovata* hydroethanolic extract (POH); dexamethasone (dexa); phosphate-buffered saline (PBS); +: inoculation of lambda carrageenan (λ-carrageenan); milligrams per kilogram (mg/kg).

**Table 3 plants-12-01814-t003:** Qualitative classification of inflammatory infiltrate after histological analysis from sections of paw edema induced by 1% λ-carrageenan in BALB/c mice, after 4 h from treatment with hydroethanolic extract of *Passovia ovata* or dexamethasone.

Group	λ-Carrageenan	Dose (mg/kg)	Classification of Inflammatory Infiltrate
Normal	PBS	PBS	Discreet
Control	+	PBS	Intense
POH	+	125	Intense
	+	250	Intense
	+	500	Moderate
Dexa	+	5	Discreet

Abbreviations: *P. ovata* hydroethanolic extract (POH); dexamethasone (dexa); phosphate-buffered saline (PBS); lambda carrageenan (λ-carrageenan); milligrams per kilogram (mg/kg).

**Table 4 plants-12-01814-t004:** Hematological parameters of BALB/c mice after 4 h of peritonitis induced by λ-carrageenan 1% administered by gavage with 500 or 250 or 125 mg/kg of *Passovia ovata* hydroalcoholic extract or dexamethasone at 5 mg/kg.

Group	λ-Carrageenan	Dose (mg/kg)	RBC (Million/mm^3^)	HGB (g/dL)	HcT (%)	MCV (fm^3^)	MHC (pg)	MCHC (g/dL)	WBC (mil/mm^3^)	Platelet Count (mil/mm^3^)
Normal	PBS	PBS	9.27 ± 0.17	16.1 ± 0.31	37.46 ± 0.5	40.4 ± 1.15	17.26 ± 0.15	42.83 ± 1.15	9.91 ± 1.45	259 ± 14.1
Control	+	PBS	9.78 ± 0.13	17.6 ± 0.21	40.25 ± 1.02	41.17 ± 1.17	18.0 ± 0.33	43.77 ± 0.57	7.68 ± 0.97	122.25 ± 6.8
POH	+	125	9.49 ± 0.19	16.4 ± 0.52	37.86 ± 0.51	39.9 ± 0.3	17.26 ± 0.2	43.36 ± 0.77	10.76 ± 4.04	155.66 ± 7.6
	+	250	9.69 ± 0.19	16.92 ± 0.71	38.66 ± 1.29	39.86 ± 0.55	17.44 ± 0.4	43.74 ± 0.43	10.06 ± 1.78	158.2 ± 6.1
	+	500	9.42 ± 0.37	16.20 ± 0.86	37.74 ± 1.75	40.06 ± 0.57	17.18 ± 0.34	42.9 ± 0.72	10.39 ± 2.69	141 ± 25.8
Dexa	+	5	9.64 ± 0.31	17.03 ± 0.89	39.03 ± 1.5	40.46 ± 0.25	17.63 ± 0.37	43.56 ± 0.77	11.32 ± 4.51	158.3 ± 36.01

Data represent mean ± standard deviation of experiment performed at least in triplicate. Abbreviations: *P. ovata* hydroalcoholic extract (POH); dexamethasone (dexa); red blood cell count (RBC); hemoglobin (HGB); hematocrit (Hct); mean corpuscular volume (MCV); mean corpuscular hemoglobin (MCH); mean corpuscular hemoglobin concentration (MCHC); white blood cell count (WBC). The sign + represents group submitted to λ-carrageenan inoculation.

## Data Availability

Data is contained within the article.

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
