# Peer review of "Effects of Passovia ovata Mistletoe on Pro-Inflammatory Markers In Vitro and In Vivo"

_plants, 2023, doi:10.3390/plants12091814_

Round 1

Reviewer 1 Report

The article entitled "Anti-inflammatory effects of Hydroethanolic Leaft Extract drom Passovia ovata mistletoe" is interesting. However, it needs to be expanded for publication in Plants Journal.
- The introduction and meterial and methods are well explained.
- Table 1 should indicate the amount of each secondary metabolite found in the analysis of plant extracts.
- The viability of the cells with the MTT method for plant extracts is not the most appropriate because it produces false positives.
- The authors should explain why they first treat the cells and then induce inflammation with LPS.
- The same scheme is followed for the study with animals. This protocol that they have followed in the investigation should be better explained.
- The reduction of nitrites and cytokines by action of plant extracts should be confirmed with western blot.
The work is well designed but needs to be completed with additional experiments.

Author Response

Reviewer #1: The article entitled "Anti-inflammatory effects of Hydroethanolic Leaft Extract from Passovia ovata mistletoe" is interesting. However, it needs to be expanded for publication in Plants Journal.

  • Table 1 should indicate the amount of each secondary metabolite found in the analysis of plant extracts

Answer: We added the amount by placing: (−) absence; (+) poor; (++) moderate; (+++) abundant.

  • The viability of the cells with the MTT method for plant extracts is not the most appropriate because it produces false positives.

Answer: Our group had good results using the MTT methodology for plant extracts in tests performed previously (for example, see doi: 10.1371/journal.pone.0225275; doi: 10.1016/j.jksus.2022.102021). The main problem associated with MTT in plants is the influence of extract color on spectrophotometric readings, since MTT is a colorimetric assay. To eliminate this problem, our research group always includes wells with each plant concentration in triplicate, which are used as a blank for each respective concentration.

  • The authors should explain why they first treat the cells and then induce inflammation with LPS.

Answer: The in vitro assay with macrophages is often standardized to previous treatment with the compound and subsequent stimulation with LPS to allow a previous contact of the cells with the compound and a result that better demonstrates the therapeutic effects of the plant. Several other works already published followed this methodology (ex: see doi: 10.1590 /fst. 15918; doi: /10.1155/2020/3164239).

  • The same scheme is followed for the study with animals. This protocol that they have followed in the investigation should be better explained.

Answer: The treatment 1 hour before the stimulus with carrageenan was recommended for paw edema by Winter et al. (1962). Oral treatment with the plant prior to stimulation with carrageenan, aims to make the treatment bioavailable for the animal considering the total treatment time (5 hours) and the peak of edema formation, which can range from 3 to 5 hours after stimulation with carrageenan, considering that for routine drug testing the peak is considered after 3 hours by Winter et al. (1962). Prior administration of the compound ensures a more effective demonstration of its potential within the short experimental period for this assay, which is also standardized for peritonitis model.

  • The reduction of nitrites and cytokines by action of plant extracts should be confirmed with western blot.

Answer: ELISA and western blots are both immunoassays that can be used in this type of research, however, ELISA has several advantages such as providing a quantitative/qualitative analysis with high efficiency, specificity and sensitivity for studying cytokines in vitro or in vivo as Chiswick et al., (2012) emphasizes (doi: 10.1007/978-1-61779-527-5_2).

Reviewer 2 Report

The article “Anti-inflammatory effects of Hydroethanolic Leaf Extract from Passovia ovata mistletoe” showed that the hydroethanolic extract of P. ovata leaves is rich in different chemical compounds with a high concentration of flavonoids and that has an anti-inflammatory effect in vitro and in vivo. The work is interesting and showed that this vegetal product can be used in the future.

Author Response

Thank you for your comments regarding our paper. We appreciate the time and effort that you and the reviewers have dedicated to providing your valuable feedback on our manuscript.

Reviewer 3 Report

In this study, the authors investigated potential anti-inflammatory effects  of  phytochemicals isolated from the mistletoe (Passovia ovata) in vitro and in vivo. In vitro, the isolated extract was tested regarding its cytotoxic potential. Moreover, potential inti-inflammatory effects were  studied in rats, in the context of cytokine expression, inflammatory cell infiltration and the generation of nitrite in vivo. IN this context, the authors found, amongst other things, that in the utilized “peritonitis model, POH did not influence cytokines levels or cell counts” and concluded that “these results demonstrate that POH has prominent anti-inflammatory effects in vitro and in vivo”.

The issue studied here seems relevant and interesting. However, this study is not well done. Several data substantiating major author statements, such as results of cytotoxicity assessment are missing. Furthermore, various data show inconclusive results regarding the potential (anti-)inflammatory effects of POH in vitro vs in vivo. In general, the experimental setting should be revised in this regard. Beside this, the study exhibits various methodological weaknesses, such as missing information regarding materials and methods and provider information. Moreover, the text exhibits different shortcomings, e.g. regarding grammar, style and phrasing.

Therefore, the study design and data interpretation have to be revised carefully. Moreover, additional experiments have to be done and necessary controls have to be included in this regard to improve the scientific quality of the current manuscript version.

Major comments:

1.    Introduction: The authors stated in line 41 – 46 that “cells release pro-inflammatory cytokines, vasoactive amines, … demands a balance between neutralization of an agent and control to heal and restore the damaged area … critical point in maintaining … involved in … cardiovascular, autoimmune, neurodegenerative and cancer [4]. In this regard, the term “diseases is missing post “cardiovascular”. Moreover, also the expression and role of factors involved in secondary hemostasis and cell signaling should be discussed in the context of inflammation and pathophysiological responses, as shown by others (e.g. see doi: 10.1161/CIRCRESAHA.108.183905; doi: 10.1253/circj.cj-99-0225;  and so on). Please also include corresponding references.

2.    Methods: Please include complete purchaser information for all utilized materials and methods, such as “ethanol” (l. 71), “rotary evaporator” (l. 76), “methanol”, “sonicator”, “Photodiode-Array Detection” (l. 81-84), “96-well plate” (l. 96), the used detection assays for anthraquinones, sapo-91 nins, tannins, anthocyanidins, chalcones, leucoanthocyanidin, catechins, flavanones, flavones, flavanols, xanthones, aurones, flavonols and coumarins  as well as the plate reader used here (lines 95-136) etc. (including name, city, state and country). The same applies to: NaOH, gelatin, filter paper, potassium hydroxide, the source of ultraviolet light, the fluorescence reader (e.g. see l. 114), HCl, aluminum chloride, the spectrophotometer (l. 133), quercetin, the MTT assay (see l. 149), the light microscope (l. 210), toluidine blue, the ELISA for cytokines IL-1β and IL-6 (l. 224), assay and materials for the Griess Assay (l. 228-238) and so on. Moreover, in line 201, different information is missing for “Schnelltaster dial gauge caliper 201 (Kröplin GRBH)” and “ketamine (100 mg/kg, Syntec, BRA) and xylazine (20 mg/kg, Syntec, BRA)” (e.g. city etc.).

3.    Methods: Include correct named and provider information for all utilized software tools (e.g. “Graph Pad” Prims and others (including version etc.).

4.    Results: In line 258 ff the authors describe the results of the MTT assay. However, the corresponding data is missing. Thus, please include a figure showing the obtained results including information about dosing, experimental repeats, controls and so on. Moreover, this was indicated in the context of the characterization of the generation of cytokines etc. in Figure 2. (Here, the units are unclear. This is quite confusing.) Since, significant effects were also seen at relatively high doses <”500” (and not in lower ranges), the results of cytotoxicity analysis should be included to allow more adequate interpretation of the obtained data on cytokine generation. In this context, the discussion have to be revised and adapted critically. Moreover, please indicate whether biological, experimental or technical replicates (n=3) were shown here and carefully discuss this in the text.

5.    Results (l. 334 ff): The authors stated that “A higher inflammatory infiltrate was observed in the stimulated and PBS-treated group (Figure 4B) and in the group treated with the lowest dose (125 mg/kg) of POH (Figure 4C). This indicates that there is no anti-inflammatory effect of POH in vivo. Substantiating this, the author stated in the abstract that in the “peritonitis model, POH did not influence cytokines levels or cell counts”. This clearly contradicts the authors suggestion that POH mediated anti-inflammatory effects. Thus, additional experiments have to be done to elucidate whether the potential beneficial effects of POH are mediated via modulation of proliferation, apoptosis, fibrosis and so on.  These analyses are mandatory to allow a differentiated interpretation of the – so far – presented inconclusive results.

6.    Discussion and Conclusions: Please carefully revise these parts of the manuscript in the context of the comments mentioned above. Especially in the context of anti-inflammatory effects observed in vitro but not in vivo and the missing findings on other potential pathways, functions, effects (e.g. proliferation, signalling etc.).

7.    Table 1 - 3: Define all used abbreviations used (e.g. “NaOH”, “Dexa” etc.) as done in Table 4.

8.    Figure 2: Define all used abbreviations used (e.g. “NaNO2”, “LPS” etc.). Moreover, check syntax and phrasing in “#p<0.01 when compared to the untreated and unstimulated group” (maybe this should be changed to “comparing the untreated and unstimulated group”. Please indicate the units for dosing of “P. ovata”. Moreover, please indicate whether biological, experimental or technical replicates (n=3) were shown here.

9.    Figure 3 and 6: Define all used abbreviations used. Moreover, please indicate whether biological, experimental or technical replicates (n=3) were shown.

10. Figure 5: Define all used abbreviations used. Please indicate the units for dosing of “P. ovata” etc. Furthermore, please indicate whether biological, experimental or technical replicates (n=3) were shown.

11.  “References”: Please check and correct all included references and use a uniform style. In this context, various mistakes were found (e.g. alternating between full journal names vs abbreviated names, missing dots or commas, small vs capital letters, full vs abbrev. page numbering and so on (e.g. see ref. 2 vs 3, 6 vs 7, 18 vs 20 etc.).

Minor comments:

1.    Abstract: Define abbreviation in the abstract, when used for the first time (e.g. see “MTT, LPS, QE, IL-1β, IL-6 and TNF-α”). And correct the term “cytokines levels” to “cytokine levels”.

2.    In general, English style, phrasing, and grammar have to be revised carefully. Beside the minor points mentioned above there are various other mistakes. The thousands separator is missing in numbers, such as “1000” (e.g. see l. 152). Please delete the hyphen in the term “1-hour” in l. 165. Sometimes space characters are missing or too much (e.g. see l 170 and 194: “Louis,MO, USA” and “5mg/kg” vs “(6) : animals” (l. 196) etc.). Correct the term “In Vivo” regarding the correct usage of small vs capital letters (e.g. l. 176). Please use a uniform style (full vs abbrev. name) for units indicating periods, such as “hour” vs “h” and “minute” vs “min”(see: l. 84, 153, 154 vs 181, 249 and so on). Define all used abbreviations when used for the first time in the text (e.g. see: "UEMA", “MTT”, “DMSO”, “PBS” (l. 191 vs 199), “ANOVA” and “LPS” etc.). Use a uniform style for terms like "p < 0.05" vs "p<0.01" with or without space characters (e.g. see l. 243 vs 261 vs 264 vs 285 etc.). Use a uniform style for terms including “anti” “pro” and/or “sub” (with or without hyphen), such as "anti-inflammatory " vs "antitumor" and “antimicrobial [21-23], antiviral, antioxidant, and anti-inflammatory action” (see l. 334), “sub-plantar” (l. 199) vs “subcutaneous” (l. 282) etc.). It is recommended to consult a native English speaker and to make use or editorial services offered by the journal.

Author Response

In this study, the authors investigated potential anti-inflammatory effects of phytochemicals isolated from the mistletoe (Passovia ovata) in vitro and in vivo. In vitro, the isolated extract was tested regarding its cytotoxic potential. Moreover, potential inti-inflammatory effects were studied in rats, in the context of cytokine expression, inflammatory cell infiltration and the generation of nitrite in vivo. IN this context, the authors found, amongst other things, that in the utilized “peritonitis model, POH did not influence cytokines levels or cell counts” and concluded that “these results demonstrate that POH has prominent anti-inflammatory effects in vitro and in vivo”.

The issue studied here seems relevant and interesting. However, this study is not well done. Several data substantiating major author statements, such as results of cytotoxicity assessment are missing. Furthermore, various data show inconclusive results regarding the potential (anti-)inflammatory effects of POH in vitro vs in vivo. In general, the experimental setting should be revised in this regard. Beside this, the study exhibits various methodological weaknesses, such as missing information regarding materials and methods and provider information. Moreover, the text exhibits different shortcomings, e.g. regarding grammar, style and phrasing. Therefore, the study design and data interpretation have to be revised carefully. Moreover, additional experiments have to be done and necessary controls have to be included in this regard to improve the scientific quality of the current manuscript version.

  1. Introduction: The authors stated in line 41 – 46 that “cells release pro-inflammatory cytokines, vasoactive amines, … demands a balance between neutralization of an agent and control to heal and restore the damaged area … critical point in maintaining … involved in … cardiovascular, autoimmune, neurodegenerative and cancer [4]. In this regard, the term “diseases is missing post “cardiovascular”. Moreover, also the expression and role of factors involved in secondary hemostasis and cell signaling should be discussed in the context of inflammation and pathophysiological responses, as shown by others (e.g. see doi: 10.1161/CIRCRESAHA.108.183905; doi: 10.1253/circj.cj-99-0225; doi: 10.1111/j.1755-5922.2010.00206.x; doi: 10.1016/j.tcm.2011.08.001 and so on). Please also include corresponding references.

Answer: Thanks for your comments about our manuscript. We include the term “diseases” after “cardiovascular” (l.50-51). We also include the role of pro-inflammatory cytokines in the pathophysiological context.

  1. Methods: Please include complete purchaser information for all utilized materials and methods, such as “ethanol” (l. 71), “rotary evaporator” (l. 76), “methanol”, “sonicator”, “Photodiode-Array Detection” (l. 81-84), “96-well plate” (l. 96), the used detection assays for anthraquinones, sapo-91 nins, tannins, anthocyanidins, chalcones, leucoanthocyanidin, catechins, flavanones, flavones, flavanols, xanthones, aurones, flavonols and coumarins as well as the plate reader used here (lines 95-136) etc. (including name, city, state and country). The same applies to: NaOH, gelatin, filter paper, potassium hydroxide, the source of ultraviolet light, the fluorescence reader (e.g. see l. 114), HCl, aluminum chloride, the spectrophotometer (l. 133), quercetin, the MTT assay (see l. 149), the light microscope (l. 210), toluidine blue, the ELISA for cytokines IL-1β and IL-6 (l. 224), assay and materials for the Griess Assay (l. 228-238) and so on. Moreover, in line 201, different information is missing for “Schnelltaster dial gauge caliper 201 (Kröplin GRBH)” and “ketamine (100 mg/kg, Syntec, BRA) and xylazine (20 mg/kg, Syntec, BRA)” (e.g. city etc.).

Answer: All information about the materials and methods was included, including the name, city, state and country of the manufacturers of the inputs used in the experiments (such as l. 85-87; l. 92-96; l.107-109; l.113; l.119-120; l.125-129; l.135-136; l.149-154; l.161; l.171-172; l.174-176; l.181; l.187; l.225-226; l.232-237; l.240; l.242; l.251; l.256-269; l.274).

  1. Methods: Include correct named and provider information for all utilized software tools (e.g. “Graph Pad” Prims and others (including version etc.).

Answer: All information were included (l.274).

  1. Results: In line 258 ff the authors describe the results of the MTT assay. However, the corresponding data is missing. Thus, please include a figure showing the obtained results including information about dosing, experimental repeats, controls and so on. Moreover, this was indicated in the context of the characterization of the generation of cytokines etc. in Figure 2. (Here, the units are unclear. This is quite confusing.) Since, significant effects were also seen at relatively high doses <”500” (and not in lower ranges), the results of cytotoxicity analysis should be included to allow more adequate interpretation of the obtained data on cytokine generation. In this context, the discussion have to be revised and adapted critically. Moreover, please indicate whether biological, experimental or technical replicates (n=3) were shown here and carefully discuss this in the text.

Answer: We have added a figure (Figure 2) that shows the cytotoxicity assay, indicating the percentage of cell viability and showing the replicates. We also revised the discussion.

  1. Results (l. 334 ff): The authors stated that “A higher inflammatory infiltrate was observed in the stimulated and PBS-treated group (Figure 4B) and in the group treated with the lowest dose (125 mg/kg) of POH (Figure 4C). This indicates that there is no anti-inflammatory effect of POH in vivo. Substantiating this, the author stated in the abstract that in the “peritonitis model, POH did not influence cytokines levels or cell counts”. This clearly contradicts the authors suggestion that POH mediated anti-inflammatory effects. Thus, additional experiments have to be done to elucidate whether the potential beneficial effects of POH are mediated via modulation of proliferation, apoptosis, fibrosis and so on. These analyses are mandatory to allow a differentiated interpretation of the – so far – presented inconclusive results.

Answer: The discussion and conclusions have been rewritten. We emphasize that the plant used in our experiments has an immunomodulatory effect by influencing, in vitro, the dosage of cytokines and nitrite and can reduce edema in vivo, all of which are considered pro-inflammatory markers. We also changed the title of the article to "Effects of Passovia ovata mistetloe on pro-inflammatory markers in vitro and in vivo", which better represents the results found in this work.

  1. Discussion and Conclusions: Please carefully revise these parts of the manuscript in the context of the comments mentioned above. Especially in the context of anti-inflammatory effects observed in vitro but not in vivo and the missing findings on other potential pathways, functions, effects (e.g. proliferation, signalling etc.).

Answer: We reviewed the discussion and conclusions of the manuscript, emphasizing that the observed effects were in the inhibition of pro-inflammatory cytokines in the in vitro experiment and in the inhibition of edema formation in vivo. We emphasize the need for further studies to clarify the mechanisms of action of the extract and its real scope in the inflammatory process.

  1. Table 1 - 3: Define all used abbreviations used (e.g. “NaOH”, “Dexa” etc.) as done in Table 4.

Answer: We included the abbreviations used for all tables.

  1. Figure 2: Define all used abbreviations used (e.g. “NaNO2”, “LPS” etc.). Moreover, check syntax and phrasing in “#p<0.01 when compared to the untreated and unstimulated group” (maybe this should be changed to “comparing the untreated and unstimulated group”. Please indicate the units for dosing of “P. ovata”. Moreover, please indicate whether biological, experimental or technical replicates (n=3) were shown here.

Answer: We included the abbreviations used, corrected the syntax and phrasing of the mentioned phrase, and indicated the units for dosing and the number of replicates for Figure 2 (which now is Figure 3).

  1. Figure 3 and 6: Define all used abbreviations used. Moreover, please indicate whether biological, experimental or technical replicates (n=3) were shown.

Answer: We corrected and included the abbreviations used, the dosage units and the number of replicates for Figure 3 and 6 (which now are Figure 4 and 5).

  1. Figure 5: Define all used abbreviations used. Please indicate the units for dosing of “P. ovata” etc. Furthermore, please indicate whether biological, experimental or technical replicates (n=3) were shown.

Answer: We corrected and included the abbreviations used, the dosage units and the number of replicates for Figure 5 (which now is Figure 6).

  1. “References”: Please check and correct all included references and use a uniform style. In this context, various mistakes were found (e.g. alternating between full journal names vs abbreviated names, missing dots or commas, small vs capital letters, full vs abbrev. page numbering and so on (e.g. see ref. 2 vs 3, 6 vs 7, 18 vs 20 etc.).

Answer: The format and reference style were altered to a uniform style.

Round 2

Reviewer 1 Report

The article has been improved and the authors have satisfactorily answered the questions presented. Therefore, the article could be accepted in the present form.

Reviewer 3 Report

The reviewer comments were not adequately addressed.